# The Role of Tenascin-C on the Structural Plasticity of Perineuronal Nets and Synaptic Expression in the Hippocampus of Male Mice

**DOI:** 10.3390/biom14040508

**Published:** 2024-04-22

**Authors:** Ana Jakovljević, Vera Stamenković, Joko Poleksić, Mohammad I. K. Hamad, Gebhard Reiss, Igor Jakovcevski, Pavle R. Andjus

**Affiliations:** 1Center for Laser Microscopy, Institute for Physiology and Biochemistry “Jean Giaja”, Faculty of Biology, University of Belgrade, 11000 Belgrade, Serbia; ana.jakovljevic@bio.bg.ac.rs; 2Center for Integrative Brain Research, Seattle Children’s Research Institute, 1900 9th Ave, Seattle, WA 98125, USA; vera.stamenkovic@seattlechildrens.org; 3Institute of Anatomy “Niko Miljanic”, School of Medicine, University of Belgrade, 11000 Belgrade, Serbia; joko.poleksic@med.bg.ac.rs; 4Department of Anatomy, College of Medicine and Health Sciences, United Arab Emirates University, Al Ain P.O. Box 15551, United Arab Emirates; m.hamad@uaeu.ac.ae; 5Institut für Anatomie und Klinische Morphologie, Universität Witten/Herdecke, 58455 Witten, Germany; gebhard.reiss@uni-wh.de

**Keywords:** tenascin-C, perineuronal nets, enriched environment, hippocampus, synaptic plasticity

## Abstract

Neuronal plasticity is a crucial mechanism for an adapting nervous system to change. It is shown to be regulated by perineuronal nets (PNNs), the condensed forms of the extracellular matrix (ECM) around neuronal bodies. By assessing the changes in the number, intensity, and structure of PNNs, the ultrastructure of the PNN mesh, and the expression of inhibitory and excitatory synaptic inputs on these neurons, we aimed to clarify the role of an ECM glycoprotein, tenascin-C (TnC), in the dorsal hippocampus. To enhance neuronal plasticity, TnC-deficient (TnC-/-) and wild-type (TnC+/+) young adult male mice were reared in an enriched environment (EE) for 8 weeks. Deletion of TnC in TnC-/- mice showed an ultrastructural reduction of the PNN mesh and an increased inhibitory input in the dentate gyrus (DG), and an increase in the number of PNNs with a rise in the inhibitory input in the CA2 region. EE induced an increased inhibitory input in the CA2, CA3, and DG regions; in DG, the change was also followed by an increased intensity of PNNs. No changes in PNNs or synaptic expression were found in the CA1 region. We conclude that the DG and CA2 regions emerged as focal points of alterations in PNNs and synaptogenesis with EE as mediated by TnC.

## 1. Introduction

Perineuronal nets (PNNs) represent perhaps the most captivating and intricate formations of the central nervous system’s extracellular matrix (ECM) [1]. These specialized structures, composed of a complex interplay of ECM proteins and carbohydrates [2], can enwrap inhibitory as well as excitatory neurons [3,4]. PNNs form an interlaced structure around neurons which is characterized by a matrix rich in hyaluronic acid [5], chondroitin sulfate proteoglycans (CSPG), such as aggrecan, brevican, neurocan, and phosphacan [6], and the glycoproteins tenascin-R and C [7,8,9]. This complex architecture serves to create a molecular scaffold around neurons that influences the surrounding microenvironment and interacts with different neuronal cell types.

PNNs are found primarily in brain regions associated with higher-order cognitive functions like emotion processing, learning, and memory, such as the neocortex [10], the amygdala [11], and the cerebellum [9,12]. One of the brain regions where PNNs are prominent and functionally important is the hippocampus, where they particularly surround interneurons [3,13,14]. Interneurons in the hippocampus play a crucial role in regulating the excitability of pyramidal neurons, which are essential for memory formation and spatial navigation [15]. The presence of PNNs in the brain implies several key functions, such as structural support and stability of synapses [16], preservation of the electrophysiological properties of the neurons they surround [17,18,19], regulation of the critical period during brain development [20], and protection from excitotoxicity by acting as a buffering system for different ions [21]. In addition, PNNs control synaptic plasticity by inhibiting the induction of LTP in specific circuits [13], and regulate learning and memory by stabilizing inhibitory interneurons that maintain the balance between excitation and inhibition [22].

PNNs have been reported to be altered in conditions such as early life stress [23], schizophrenia [24], Alzheimer’s disease (AD) [25], ischemia [26,27], and Rett syndrome [28]. In fact, the dysfunction or disruption of PNNs in the hippocampus has been associated with the modulation of memory [29], highlighting their essential role in higher cognitive functions. 

Enriched environment (EE), a behavioral protocol for inducing brain plasticity that includes a combination of sensory, cognitive, and motor stimulation [30], has been shown to influence PNNs in various brain regions. For example, PNNs were found to be degraded in the cerebellum of adult mice after housing in EE for one [31] and two months [9], and also in the somatosensory cortex after experimental stroke following EE for 9 weeks [32]. Contrary to those findings, EE for the duration of 3 months led to an increased number of PNNs in the visual cortex of monocular-deprived animals [33] and an increase in PNN staining in the CA2 region of the hippocampus after one month of EE [3] while longer exposure to EE, such as 6 months, had beneficial effects on the memory of AD animal models by rescuing the number of PNNs on parvalbumin-positive (PV+) neurons [34].

TnC is an ECM glycoprotein that plays a crucial role in several developmental and functional processes in the nervous system [35,36]. It is involved in neuronal and glial cell development, migration, and neurogenesis [37,38,39,40], proliferation and regeneration [41], and neurite growth [42]. In pathological conditions, TnC is instrumental in spinal cord regeneration and glial scar formation [43,44,45], as well as in inflammatory processes in experimental autoimmune encephalomyelitis and ischemia [46,47,48]. Expression of TnC is highly dynamic during development, when physiological levels of mRNA and protein expression of TnC significantly decrease as the organism matures [37]. TnC is found in the hippocampus of developing rats [37] and in the forebrain of domestic chickens [49], and is detectable in all layers of the developing mouse cerebral cortex [50]. However, certain structures, such as the somatosensory cortex [7], hypothalamus [51], cerebellum [9], and the pyramidal and granular cell layer of the hippocampus [37], continue to show significant levels of TnC expression into adulthood. The first behavioral study using TnC-deficient (TnC-/-) mice examined their ability to encode spatial information, and found that their performance in the water maze was unaltered [52]. However, in our previous study, we showed impaired spatial learning capabilities in TnC-deficient mice, using the Morris water maze test, albeit detectable only after being reinforced by housing in an enriched environment [53]. Morellini and Schachner [54] further addressed hippocampal functions of TnC-/- mice by showing impaired extinction of the conditional fear response and enhanced novelty-induced activity in male mice deficient in TnC. We also reported enhanced novelty-induced locomotion and exploration activity on TnC-deficient mice, which was stabilized by EE [53]. Also, TnC mutants exhibited spontaneous nocturnal hyperactivity, poor sensory coordination, and low swimming velocity [53]. In the fear conditioning paradigm, TnC-/- mice had normal learning and memory capabilities, but their extinction of conditional fear response was impaired [55]. The TnC molecule has been particularly studied in the context of learning and memory, where it has demonstrated an important function in synaptic plasticity [9,52,56,57,58,59,60,61,62]. 

TnC has been shown to interact with the major components of PNNs, aggrecan [63], neurocan, and phosphacan [64,65]. An initial study of PNN expression in the hippocampus of TnC-deficient mice showed that, in young adults (4–6 weeks old), the presence of PNNs remains unaffected [52]. Similarly, expression of TnC in the cortex of adult mice was found to be generally low in the regions with abundant PNNs, and the appearance of PNNs in TnC-/- mice closely resembled that of wild-type mice [38]. Interestingly, we have also previously shown that PNNs in the cerebellum of 7-week-old TnC-/- mice had the same appearance as in wild-type mice, but they were less condensed 4 weeks later, indicating that developmental maturation and stabilization of PNNs in the cerebellum might be altered in the absence of TnC [9]. Moreover, recent studies indicated a delay in the maturation of PNNs in the CA2 region of the hippocampus in mice lacking four ECM molecules (TnC, TnR, neurocan, and brevican), although the specific contribution of the individual molecules is unclear [57]. 

We previously demonstrated that EE for a duration of 2 months increased the levels of TnC in the cerebellum [9]. Given that TnC expression declines with ageing, incorporating enriched environment represents a non-invasive approach to increase brain plasticity together with upregulating expression of TnC. Moreover, in conditions of increased plasticity by EE, TnC acts as a modulatory molecule for PNN expression [9]. In addition to affecting PNNs and TnC expression, EE leads to changes in the number of inhibitory and excitatory synaptic puncta in the cerebellum [9,31]. 

Thus, to investigate the role of TnC in PNN expression and in structural synaptic plasticity of the hippocampus, we reared the animals in EE and examined the number and intensity of PNNs around PV+ and PV- cells, the PNN ultrastructure, and density, size and intensity of inhibitory and excitatory projections on PNN-enwrapped neurons. In the present study, we analyzed four subfields of the hippocampus (dentate gyrus (DG), and Cornu Ammonis 1, 2, and 3 (CA1, CA2, and CA3)) in TnC+/+ and TnC-/- mice after 8 weeks of exposure to standard or enriched housing conditions starting at postnatal day 21. For the first time, we addressed the role of TnC in the expression of PNNs and synaptic markers, as well as the PNN ultrastructure in all four hippocampal subfields. We demonstrated that the depletion of TnC leads to a diversity of responses in the expression of PNNs and synaptic inputs, across hippocampal subfields, particularly emphasizing the importance of TnC in shaping the structural plasticity in the DG and CA2 regions of the hippocampus.

## 2. Materials and Methods

### 2.1. Animal Model

In this study, wild-type (TnC+/+) and tenascin-C constitutively deficient (TnC-/-) male mice inbred on the C57BL/6 background were used. The generation of TnC-/- mice and protocol for genotyping have been previously described by Evers et al. [52]. All efforts were made to minimize animal suffering and to reduce the number of animals used in this study. All experiments were carried out according to the NIH Guide for Care and Use of Laboratory Animals and were approved by the Ethics Committee of the University of Belgrade. For each experiment, the number of animals was 3–4 per experimental group, based on our previous experience [9].

### 2.2. Enriched Environment

Enriched environment (EE) is an experimental paradigm that provides exploratory, motor, cognitive, and sensory stimulation [9,30]. It refers to housing animals in specialized housing cages (54 × 39 × 27 cm) with different objects (running wheels, climbing stairs, tunnels, platforms, and small boxes) that vary in composition, shape, and texture. In order to promote exploratory behavior, the location of the objects was changed daily, whereas the composition of the objects was changed once per week. Given that EE exposure starting from early age has been shown as very effective approach that remarkably changes brain structure and function in a non-invasive way [66], we reared TnC-/- and TnC+/+ male mice in enriched or standard conventional cages (32 × 20 × 13 cm; standard environment (SE)) beginning from postnatal day 21 (PND21). Animals were housed in groups of 4–5 mice per cage for time period of 8 weeks. All animals were maintained in a standard 12 h light/dark cycle (lights on from 06:00 to 18:00), with tap water and food available ad libitum.

### 2.3. Immunohistochemistry

After 8 weeks of EE and SE, at PND 81, all animals were anesthetized with ketamine (100 mg/kg) and xylazine (10 mg/kg) solution and transcardially perfused with 0.9% saline followed by 4% paraformaldehyde (PFA, Sigma-Aldrich, Darmstadt, Germany) in 0.1 M phosphate buffer pH 7.4. Isolated brains were post-fixed in a 4% PFA for 24 h at +4 °C, cryoprotected in 30% sucrose in 0.1 M phosphate buffer for 48 h at +4 °C, and stored at −80 °C until sectioning. Coronal sections of dorsal hippocampus 30 and 10 µm thick were cut on a freezing cryostat (Leica Instruments, Nussloch, Germany), at −25 °C for confocal and super-resolution microscopy, respectively, and collected on Superfrost Plus glass slides (Menzel Glaser, Braunschweig, Germany). In this study, we used biotinylated lectin from Wisteria floribunda agglutinin (WFA 1:100, L1516, Millipore Sigma, Burlington, MA, USA) to label the PNNs and mouse anti-parvalbumin monoclonal antibody (PV, 1:1500; MAB1572, MilliporeSigma, Burlington, MA, USA) to label the inhibitory interneurons. In addition, to visualize inhibitory and excitatory presynaptic terminals, we used rabbit anti-vesicular GABA transporter (VGAT, 1:200, 131003, Synaptic Systems, Goettingen, Germany) and guinea pig anti-vesicular glutamate transporter 1 (VGlut1, 1:600, 135304, Synaptic Systems, Goettingen, Germany) antibodies, respectively. After thawing, sections were first rehydrated in phosphate buffer saline (PBS) for 30 min. For WFA/PV and WFA/VGAT/VGlut1 immunofluorescence, antigen retrieval was performed in 10 mM sodium citrate buffer with 0.05% Tween 20, pH 6 on +75 °C for 10 min, following the protocol of Jakovljevic et al. [23]. For WFA/PV immunofluorescence non-specific binding was blocked with 5% normal donkey serum (NDS, Sigma-Aldrich, Germany) with 0.2% Triton X-100 in PBS for 1 h, whereas 20% NDS in PBS was used as a blocking solution for sections stained for inhibitory and excitatory synapses [67]. To block all endogenous biotin, biotin receptors, and streptavidin binding sites present in tissues sections stained for biotinylated, WFA alone or in combination with synaptic markers were additionally incubated in a streptavidin–biotin blocking kit (Vector Laboratories, Newark, CA, USA) according to manufacturer’s instructions. Sections were then incubated with WFA and appropriate primary antibodies overnight at 4 °C. After incubation, sections were washed in PBS (3 × 15 min) and incubated for 2 h at room temperature with appropriate secondary antibodies conjugated to a fluorophore: Alexa 488-conjugated streptavidin for WFA lectin, Alexa-555 conjugated donkey anti-mouse for PV and VGAT antibody, as well as Alexa-647 conjugated donkey anti-guinea pig for VGlut1 antibody. Secondary antibodies were purchased from Invitrogen (Waltham, MA, USA) (streptavidin AF 488 (S11223), donkey anti-mouse AF 555 (A31570), donkey anti-rabbit AF 555 (A31572)) and Jackson Immuno Research (Newmarket, UK) (donkey anti-guinea pig AF 647 (NC0292731)) and used at a dilution of 1:200. The sections were washed with PBS (4 × 15 min) and stained with the nuclear marker 4,6-diamidino-2-phenylindole (DAPI, 1:4000, Molecular Probes, Eugene, OR, USA) for 10 min. After rinsing with in PBS (3 × 10 min) slides were coverslipped using Mowiol (Sigma-Aldrich, Germany) mounting medium for confocal and 90% glycerol for super-resolution microscopy, and left to dry out overnight. Negative control staining was performed by omitting primary antibodies. Sections from all experimental groups were processed in parallel for every mentioned immunohistochemical reactions. To standardize the analysis and control for variations in hippocampal region sizes across different groups, we consistently selected the same coronal section level for each group. This specific level corresponds to the range of −1.58 to −1.94 distance to the bregma, as defined by Paxinos and Franklin [68].

### 2.4. Image Acquisition

PNNs, PV+ cells, and synaptic markers were analyzed on 3–4 successive sections from one hemisphere of the dorsal hippocampus. Image acquisition was performed on the confocal laser scanning microscope (LSM 510, Carl Zeiss, GmbH, Jena, Germany), equipped with Ar (488 nm) and HeNe (543 nm, 633 nm) lasers, using 40× (Plain Apochromat, NA = 1.3) oil immersion objective for imaging PNN and PV+ neurons and 63× (DIC, NA = 1.4) oil immersion objective for imaging synaptic markers. WFA+ and PV+ cells were imaged in the 2D acquisition mode. For imaging of PNNs and PV neurons using 40× objective, image fields were selected to be in the nearly same position for every hippocampal subfield. This approach ensured that the same regions were consistently analyzed across different animals and groups. In each animal, two images were imaged for the hippocampal subfields DG, CA1, and CA3, and due to the dimensions of the CA2 subfield, only one image per animal was taken. Triple staining of PNNs, VGAT, and VGlut1 positive synaptic markers were imaged in 3D mode (7 optical slices of 1 μm thickness). Section depth for imaging synaptic markers was up to 10 μm, based on Ippolito and Eroglu [67], where an antigen retrieval protocol was used to cleave the epitopes and longer incubation time of primary antibodies allowed better penetration of synaptic markers. For imaging PNNs and synaptic markers, 2–4 PNNs per hippocampal subregion per animal were acquired for analysis. The ultrastructure of PNNs around the neuronal soma was imaged using a super-resolution structured illumination microscope (SIM, Delta Vision OMX V4, GE Healthcare, Chicago, IL, USA) equipped with a 405 nm diode laser and 488 nm OPSL laser, in 3D mode (50–55 optical slices of 125 nm thickness) using 60× (PlanApoN, NA = 1.42) oil immersion objective, gaining overall lateral resolution of 150 nm and 300 nm in axial direction. In each animal, 5–6 images of the DG region, 9 images of the CA1 region, 5–6 images of the CA2 region, and 3–4 images of the CA3 region were acquired for analysis. For imaging PNNs ultrastructure and PNN with synaptic markers, PNNs were randomly selected for imaging, due to the smaller field of view upon high magnification objectives. Imaging settings were identical for every experimental group, allowing comparisons of pixel intensities between groups (Appendix A).

### 2.5. Image Analysis

All images were acquired and analyzed manually by an observer blind to the experimental conditions. In each experiment, the hippocampal area was divided into the dentate gyrus (DG), CA1, CA2, and CA3 subregions, and imaged and analyzed as such. The number of analyzed PNNs per hippocampal region is presented in Appendix A. We utilized WFA staining to delineate hippocampal subregions because WFA labels the extracellular matrix excluding the densely packed pyramidal and granular cell layers. Under lower magnification objectives (10× and 20×) and fluorescent excitation, we visually identified the hippocampal subregions by comparing them to the Mouse Brain Atlas [68]. Once confirmed, we focused on PNNs within the specific subregion and then adjusted the objective for higher magnification. Analysis of PNNs was not separated according to the hippocampal layers; however, most PNNs were found to be expressed in the subgranular layer of the DG region and the pyramidal layer of the CA1, CA2, and CA3 regions. Nevertheless, a future study is justified to check this distribution. Confocal images were analyzed in FIJI, an open-source image processing package based on ImageJ (ImageJ v.1.46R, NIH, Bethesda, MD, USA), whereas images acquired with super-resolution SIM microscope were analyzed using IMARIS software (IMARIS, Oxford Instruments, Abingdon, UK, version 9.2).

#### 2.5.1. Quantification of WFA and PV Signals

In each section, number and intensity of overall PNNs, PNNs around PV positive (WFA+/PV+) and PV negative (WFA+/PV-) cells, and number of PV+ cells were quantified within each subregion of the hippocampus. After subtraction of background brightness using the rolling ball radius function [69], each PNN was manually delineated as a region of interest (ROI). Intensity of PNNs was analyzed firstly by removing background staining using the rolling ball radius function [69]. Mean WFA signal pixel intensities were assessed in selected ROIs, with pixel values in the range of 0–255. Average signal intensity values per subregion of the hippocampus were used in further analysis. 

#### 2.5.2. Quantification of Density, Size and Intensity of Synaptic Puncta

Analysis of synaptic puncta was performed using the FIJI function “Analyze particles”. After subtracting the background signal, the number, size, and intensity of VGAT and VGlut1 positive synaptic puncta penetrating through the PNN mesh were quantified from ROIs delineating PNNs. Average density of synaptic puncta was calculated by dividing the number of all puncta in a ROI delineating a PNN, obtained through the “Analyze particles” tool, by the area of the ROI, expressed in μm^2^. It provided information about the density of inhibitory and excitatory synaptic puncta within the PNN. Average size of synaptic puncta was obtained with the same tool, where the size of all puncta within a ROI of PNN was measured, divided by their number, and expressed in μm^2^. The aim of measuring the size of the synaptic puncta was to give deeper insight into the puncta morphology. The same tool displayed average pixel intensity of synaptic puncta. The value of the pixels, viewed as brightness of staining, is often proportional to the amount of target synaptic protein, where higher pixel intensity suggests a higher concentration of the specific synaptic marker. For each experimental group, average synaptic density, size, and intensity were calculated for each PNN independently (not averaged per slice) and used as such for statistical analysis.

#### 2.5.3. Analysis of PNN Ultrastructure

Super-resolution images of PNNs obtained by SIM were processed in IMARIS by generating spots which corresponded to nodes in the PNN mesh. Node positions were exported from IMARIS and imported to MatLab2019. Thanks to a custom algorithm developed by Dzyubenko et al. [26], node coordinates were used to calculate 5 topological metrics for analyzing the PNN ultrastructure: number of nodes in PNN mesh, percentage of connected nodes in the mesh, average distance between nodes, average number of nodes connections, and R95, a parameter which measures maximal internode distance starting from 0.7 µm until it reaches 95% of connected nodes. Values of all topological metrics were averaged from individual PNNs for each experimental group and used in statistical analysis.

### 2.6. Statistical Analysis

In this study, genotype and housing conditions were regarded as two independent factors. As a parametric test, two-way ANOVA followed by Tukey’s post hoc test was used to determine if two independent factors interact with each other among experimental groups, and a non-parametric Kruskal–Wallis H test followed by Dunn’s post hoc test were used for data that did not follow normal distribution. Data processing and statistical analyses were performed in JASP software (Version 0.12; JASP Team, 2020). The level of statistical significance was set at *p* = 0.05. Data were visualized using R (RStudio, version 4.3.3, PBS, Boston, MA, USA) and presented as box plots. The number of analyzed PNNs per experimental group for each analysis in all hippocampal regions is specified in Appendix A.

## 3. Results

To investigate the role of TnC in structural plasticity of the dorsal hippocampus, TnC +/+ and TnC-/- mice were reared in enriched and standard environment for 8 weeks starting from PND 21. Since it is known that parvalbumin interneurons contain elaborate PNNs (PNN+PV+), we examined both number and intensity of this population, as well as remaining PNN+PV- neurons. However, here, we will be presenting only the PNN+PV- population, since in our hands no significant differences were seen in PNN+PV+ cells. Next, the role of TnC in maintaining the ultrastructure of PNN in the hippocampus was revealed using super-resolution microscopy and advanced image analysis. Finally, we measured the expression of inhibitory and excitatory projections on PNN-enwrapped neurons. 

### 3.1. TnC Shapes the Ulstrastructure of PNNs and Remodels the Excitatory and Inhibitory Puncta in DG

Rearing animals in an enriched environment (EE) induced a strong increase in WFA signal intensity from PNNs of the DG (Figure 1A,B) followed by an increase in the density of VGAT puncta (Figure 2A,B), suggesting a role of PNNs in the stabilization of synaptic inputs in wild-type mice. The effect of EE on the expression of PNNs and their influence on synaptogenesis could be mediated by TnC, since the change in PNN signal intensity was not seen here in TnC-/- mice.

The depletion of TnC (TnC-/-) was found to induce a depletion of the number of PNN nodes, confirming the role of TnC in the condensation of PNNs (Figure 3A,C). Observed changes thus indicate that TnC regulates fine granulation of the PNN mesh in DG as well as its synthesis in general. Furthermore, a role of TnC was demonstrated in modulating structural synaptic plasticity in DG, since TnC deficiency affected excitatory and inhibitory inputs that are projecting on the neuronal bodies through PNNs. Thus, upon the depletion of TnC (TnC-/- mice), the density of the inhibitory VGAT puncta was found to be significantly increased, while the intensity of excitatory VGlut1 puncta was decreased (Figure 2A,B,D). Taken together with changes in the PNN ultrastructure, this extensive synaptic remodeling points to the importance of TnC in the regulation of DG plasticity. EE can restore the PNN mesh structuring in TnC-/- mice by increasing the number of nodes (Figure 3A,C), followed by an increased coverage of VGlut1 synaptic puncta (Figure 2A,C). 

### 3.2. TnC Regulates the Number of PNNs and the Inhibitory Input in the CA2 Region

In the region of Cornu Ammonis, divided into the CA1, CA2, and CA3 subfields, the most prominent changes in the expression of PNNs and synaptic markers were reported in the CA2 region. TnC was indicated to play a role in the preservation of PNNs in the CA2 region, since a depletion of TnC (TnC-/- mice) had a significantly higher number of PNNs compared to the wild-type mice (Figure 4A,B). This result was also associated with an increased intensity of VGAT puncta (Figure 5A,B). EE reversed the number of PNNs in the CA2 region of TnC-/- mice, by decreasing their elevated number (Figure 4A,B). In the CA2 (Figure 5A,B) and CA3 regions (Figure 6A,B) of wild-type mice, EE increased the density of VGAT puncta. These data point to an interaction of TnC- and EE-induced synaptic remodeling. 

No significant changes in PNN expression were reported in the CA1 and CA3 regions upon TnC deficiency or upon exposure to enriched environment. Additionally, there was no significant difference in any of VGlut1 markers for the CA3 subfield, and interestingly, no changes in the expression of inhibitory or excitatory puncta among groups were observed in the CA1. In the DG, CA2, and CA3 regions where synaptic remodeling was reported, changes did not include alterations in the size of synaptic puncta (Appendix A). Topological metrics of the PNN mesh in hippocampal subregions CA1, CA2, and CA3 were not significantly different among groups (Appendix A). 

## 4. Discussion

The results of our study demonstrate the complex role of TnC in the structural plasticity of the hippocampus. Notably, TnC deficiency and environmental conditions had distinct effects on different hippocampal subfields. Overall, the most prominent changes in structural and synaptic plasticity were found in the DG and CA2 regions.

### 4.1. Role of TnC in the Plasticity of the Dentate Gyrus

This is the first study to investigate the PNN ultrastructure in the hippocampus of wild-type and TnC-deficient mice using SIM super-resolution microscopy. Analyzing the ultrastructure of PNNs in DG, we identified a reduced number of nodes in the PNN mesh of TnC-deficient mice. This change in fine granulation confirmed the role of TnC in maintaining the condensation state of PNNs. In the study of Dzyubenko et al. [26], SIM was used for imaging cortical PNNs after ischemia and mild hypoperfusion. This study revealed that a PNN does not simply undergo a degradation state, but that its nodes change their interlinking patterns as a reorganization strategy to maintain the balance between neuroprotection and neuroplasticity. Our findings reveal that, together with regulating PNN condensation, TnC has emerged as an important modulator of structural synaptic plasticity. Namely, in the absence of TnC, there are more inhibitory and less excitatory inputs onto neurons enwrapped by PNNs in DG. The observed outcomes are in agreement with the generally accepted notion that a less-condensed PNN allows more synaptic projections on its neuron’s body [70]. Moreover, the depletion of TnC may lead to a shift in the excitation/inhibition balance, potentially impacting the stability of neuronal networks within the DG. Thus, TnC could preserve proper synaptic expression by maintaining the ultrastructural condensation state of the PNN mesh. 

As a plasticity-inducing non-invasive stimulation, EE causes specific changes in the brain of wild-type mice. Our results show that 8 weeks of EE leads to a rise in PNN intensity and its inhibitory inputs in the DG. These findings confirm the stabilizing role of PNNs in anchoring synaptic projections [70], and suggest that synaptogenesis, reported to be increased by enriched environment [71], might occur on inhibitory projections on neurons enwrapped by the PNNs. We previously showed on the cerebellum of wild-type mice that the same duration of EE reduces the PNN intensity and the size of the inhibitory puncta [9]. Since the duration of EE and the animal strain were the same in the previous and current studies, the difference in the outcomes indicates that the PNNs in the cerebellum respond differently to EE compared to those in the hippocampus, which is in correlation with different functional roles of those two structures. Furthermore, the increase in PNN number after 9 weeks of EE was found to contribute to the recovery from the visual impairment induced by monocular deprivation [33]. This is congruent with our results in the DG and may imply that intense visual and spatial stimulation, experienced in the EE, could lead to increased synthesis of PNN components, and thus, stabilization of new synapses. PNNs main constituents, CSPG, cover the neurons of the subgranular layer of the DG, an active zone of adult neurogenesis [72]. The important finding of this study was that after two weeks of EE, the synthesis of CSPG increased, which restored the production of new granule cells. Similarly, our results demonstrated an increase in the PNN intensity after EE in the DG, and an increased expression of PNNs in the subgranular layer of DG in both wild-type and TnC-deficient mice. Additionally, residual TnC immunoreactivity was detected in the molecular layer of DG and the hilar region [52,61], in which the PNNs are most abundant. The population of PNNs that changed under the conditions of EE was not expressing parvalbumin, thus we assume that it may belong to the other population of interneurons expressed in the hippocampus, such as cholecystokinin, calretinin, vasoactive intestinal peptide, somatostatin, reelin, neuropeptide Y, or calretinin expressing interneurons [73]. 

The interaction of the TnC-deficient genotype and the EE resulted in distinct alterations of PNNs and synaptic expression in the DG. EE recovered the number of nodes in the PNN mesh and the excitatory input in TnC-deficient mice. Considering this, EE could have a compensatory effect on TnC deficiency background. On the other hand, the dynamic role of TnC in the DG might be dependent on the environmental context.

### 4.2. Role of TnC in the Plasticity of CA1-3 Regions

Within the CA regions, TnC emerges as a key player in preserving PNNs and synaptic plasticity of the CA2 region. Notably, the depletion of TnC resulted in an increase of the PNN number and an increased presence of inhibitory puncta. Simultaneous increase in the PNN number and in the inhibition input in the CA2 region may reflect a role of PNNs in anchoring new synaptic connections [70]. Since the change in the number of PNNs in CA2 region of the TnC-deficient mice originated from the cells that were not positive for PV, the pressing question is the type of neurons that PNNs in the CA2 region enwrap. Comparing with other hippocampal subfields, our results confirmed that CA2 could be distinguished by a higher concentration of PNNs [3] and relatively more PV neurons [74]. PNNs in the CA2 subfield were found to be positive for the excitatory neuronal marker PCP4, in agreement with [10]; thus, this population of PNNs could be expressed around excitatory neurons which have an important role in social memory [75]. Deficits in the formation of PNNs were reported in primary hippocampal neurons of quadruple knock-outs for four ECM molecules (TnC, TnR, brevican, and neurocan), where the percentage of neurons enwrapped with PNNs and the size of the net were reduced [76]. Other findings from the same quadruple knock-out model showed that the maturation of PNNs was significantly delayed in the CA2 subfield, but reached a normal stage after postnatal day 30 and remained unchanged in adulthood [57]. In contrast, we here report a change in PNN number in the CA2 region in young adults deficient of TnC that might point to the specific contribution of TnC to the maintenance of PNNs through adulthood. However, we acknowledge that the interactions of TnC with other molecules deleted in the quadruple knock-out present a particular challenge. 

The CA region of wild-type mice did not display any change in PNN parameters as a response to EE. However, it was shown that early exposure to EE, from neonatal age until P21 and P45, leads to an increase in PNN intensity in the CA2 [3]. The most likely explanation for the discrepancy from our results is a later introduction and different duration of EE in our research. Exposure to EE may have an influence on PNNs in the CA2 only if it occurred before the closure of the plasticity window [77]. Otherwise, PNNs may remain unaffected, while the synaptic effects could persist, since there was an increase in inhibitory inputs in the CA2 and CA3 region. Moreover, the function of TnC as a regulator of synaptic integrity becomes more pronounced in the conditions of increased neuronal stimulation induced by EE. This effect is evident in the CA2 and CA3 regions, where EE only in presence of TnC affects the PNN number in the CA2 region and increases the inhibitory input to both regions.

PNNs in the TnC-deficient murine model were firstly examined in the CA1 region by Evers et al. [52], who found no change in the PNN intensity in 7-week-old mice. Our findings are consistent with the aforementioned study, since no change in the expression of PNNs was found in the CA1 region of 11-week-old TnC-deficient mice. In addition, it was reported that the number of PV+ cells did not alter in the CA1 region of the TnC-/- mice [58], in good agreement with our results. Nonetheless, lower density of PV+ neurons and a reduced ratio of excitatory to inhibitory neurons was found in the sensory cortex of TnC-/- mice by Irintchev et al. [38], indicating the importance of TnC for normal cortical development. The lack of the effect of TnC depletion in the CA1 region may reflect the absence of involvement of this biomolecule in associative memory and spatial navigation, the primary functions of the CA1 region. This assumption is supported by the previous study that demonstrated that spatial learning and memory were unaltered in TnC-/- mice [52]. Measuring VGlut1 parameters proved to be challenging in the CA1 region, due to a high expression of this protein in the *stratum oriens* and *stratum radiatum* leading to a high background signal, and on the other hand, a low expression in the *stratum pyramidale*. Also, the distribution of PNNs varies in different hippocampal layers and PNNs can be expressed exclusively in the pyramidal layer, where the expression of VGlut1 is low and thus presented as zero density and zero intensity in the graphs. The CA3 region is the least investigated hippocampal region in the TnC-deficient model, with no previous data on PNN expression, while no changes in its expression were found in the present study.

Our results suggests that in the condition of a TnC deficiency and a stimulating environment, the plasticity changes may occur with different manifestations in different subfields of the dorsal hippocampus. Importantly, changes in synaptic expression under different genetic and environmental conditions were mostly in agreement with the changes in the expression of PNN. TnC could be specifically involved in modulating structural and synaptic plasticity within the functional networks in the DG and CA2 regions, which are responsible for pattern separation and social memory, respectively, without affecting primary functions of the CA1 and CA3 regions. Of note, in our study, we only used male mice based on comparison with previous experience with males in the TnC model [9,56]. Thus, further study of the expression and ultrastructure of PNNs with comparison of female and male mice, together with measurements of inhibition and excitation in the hippocampus of TnC-deficient mice, is warranted.

## 5. Conclusions

This study presents, for the first time, the effects of TnC deletion on PNNs and synapses in the dorsal hippocampus. The most prominent effects of TnC deficiency are seen in the DG and CA2 subfields, where TnC is shown to be instrumental in decreasing synaptic plasticity by keeping the integrity of PNNs. An enriched environment may have additional effect on PNNs and synaptic marker expression, as it increases plasticity, which could rescue some of the changes caused by TnC deficiency. Differences in the effects of TnC deficiency that we observed in different hippocampal subfields indicate that the role of TnC in the development and/or maintenance of PNNs and synapses might be important for pattern separation and social memory. It will be interesting to look further into structural and behavioral mechanisms behind the temporal and spatial dynamics of TnC functions.

## Figures and Tables

**Figure 1 biomolecules-14-00508-f001:**
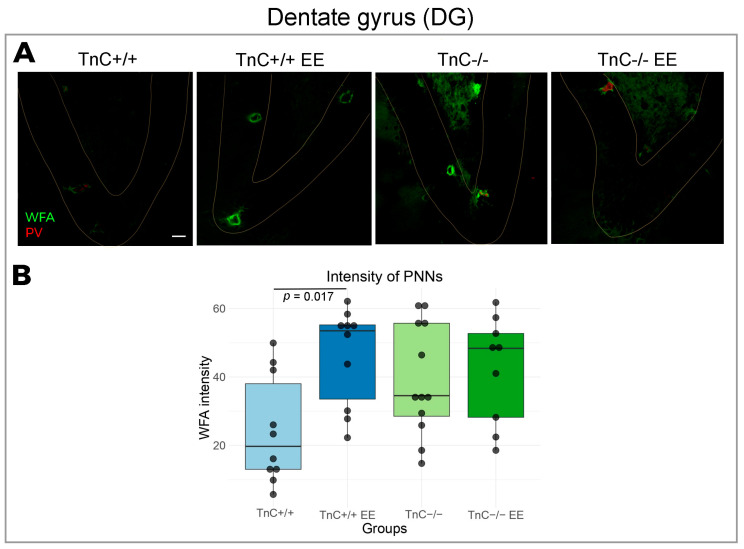
Expression of PNNs in the region of DG in the hippocampus and the role of TnC and exposure to EE. (**A**) Representative confocal images showing double immunofluorescent staining for PNN marker WFA (green) and parvalbumin (PV). Granular layer of the DG is delineated with light yellow line. Here and elsewhere, we refer to wild-type mice (TnC+/+), wild-type mice housed in an enriched environment (TnC+/+ EE), TnC-deficient mice (TnC-/-), and TnC-deficient mice housed in an enriched environment (TnC-/- EE). (**B**) Box plot depicting the median intensity of PNNs (4 animals per experimental group, 4 brain section per animal); whiskers extending from the boxes indicate the range of data within 1.5 times the interquartile range. A significant rise was revealed in the PNN intensity after EE in WT (TnC+/+) mice (two-way ANOVA, *p* = 0.017). Scale bar (**A**): 50 μm.

**Figure 2 biomolecules-14-00508-f002:**
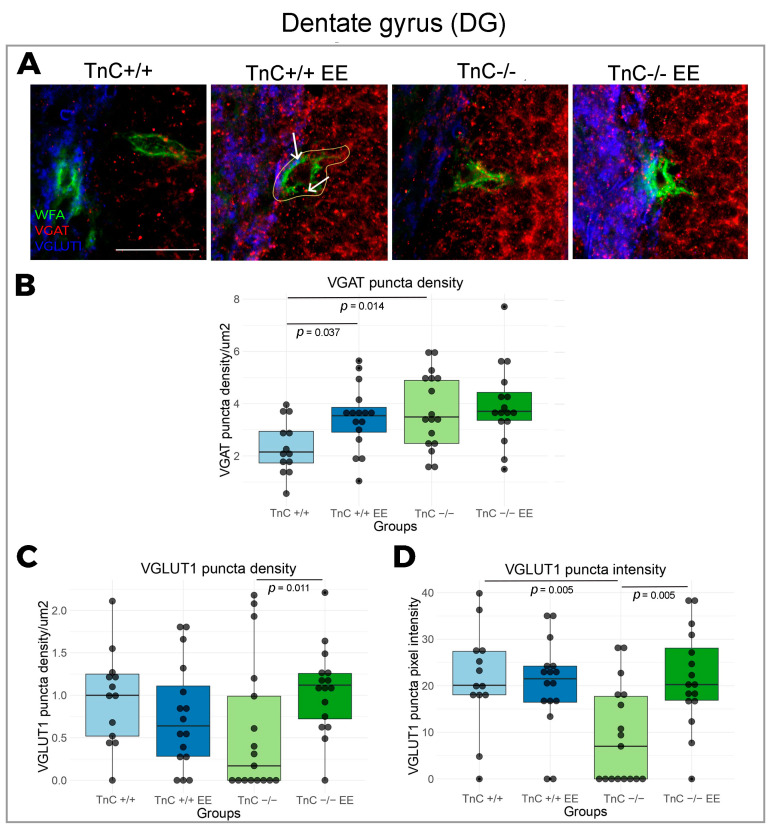
Expression of inhibitory and excitatory synapses in the DG of the hippocampus and the role of TnC and exposure to EE. (**A**) Representative 3D confocal images show triple immunofluorescent staining for the PNN marker WFA (green), the inhibitory synapse marker VGAT (red) and the excitatory synapse marker VGlut1 (blue). White arrows point to the VGlut1 and VGAT synaptic puncta quantified within the defined region of interest (ROI) that delineates the PNN (yellow line). (**B**–**D**) Box plots depicting the density and intensity of inhibitory and excitatory synapses that are penetrating the PNN (3–4 animals per experimental group, 3 brain section per animal); whiskers extending from the boxes indicate the range of data within 1.5 times the interquartile range. Significant differences were reveled in (**B**) VGAT density with EE and upon TnC depletion (Kruskal–Wallis, *p* = 0.037; *p* = 0.014, respectively), (**C**) VGlut1 density with EE on TnC depleted (TnC-/-) mice (Kruskal–Wallis, *p* = 0.011), and (**D**) VGlut1 intensity upon TnC depletion (TnC-/-) and with EE in these mice (Kruskal–Wallis, both *p* = 0.005). Scale bar (**A**): 50 μm.

**Figure 3 biomolecules-14-00508-f003:**
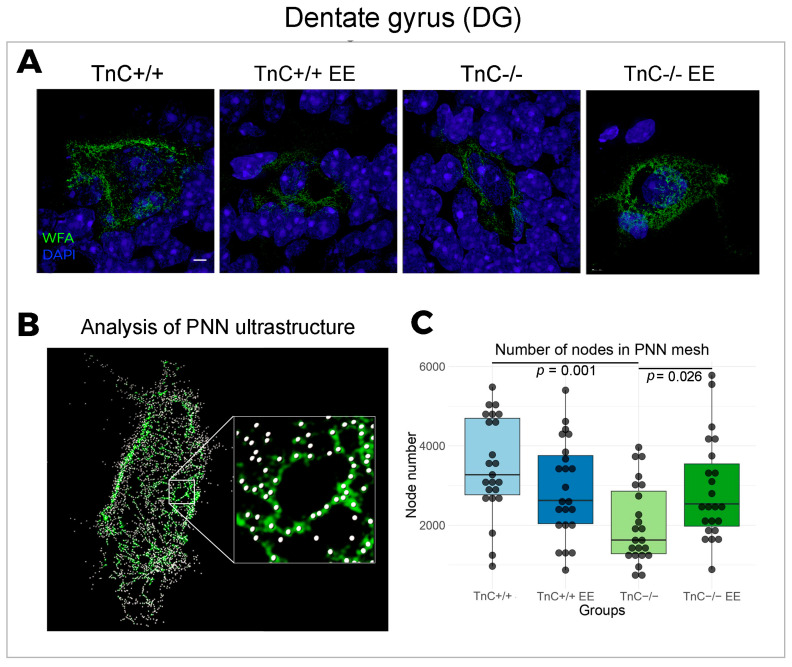
The ultrastructure of PNN in the region of DG in the hippocampus and the role of TnC and exposure to EE. (**A**) Representative 3D super-resolution SIM images show double immunofluorescent staining for the PNN marker WFA (green) and for DAPI (blue). (**B**) Zoomed-in detail of the PNN mesh, showing the nodes used for the topological analysis of the PNN ultrastructure. (**C**) Box plot depicting the median number of nodes in the PNN mesh (4 animals per experimental group, 3 brain section per animal); whiskers extending from the box indicate the range of data within 1.5 times the interquartile range. Significant differences were revealed in the number of nodes in the PNN mesh after TnC depletion (TnC-/-) and in these mice with exposure to EE (Kruskal–Wallis, *p* = 0.001; *p* = 0.026, respectively). Scale bar (**A**): 3 μm.

**Figure 4 biomolecules-14-00508-f004:**
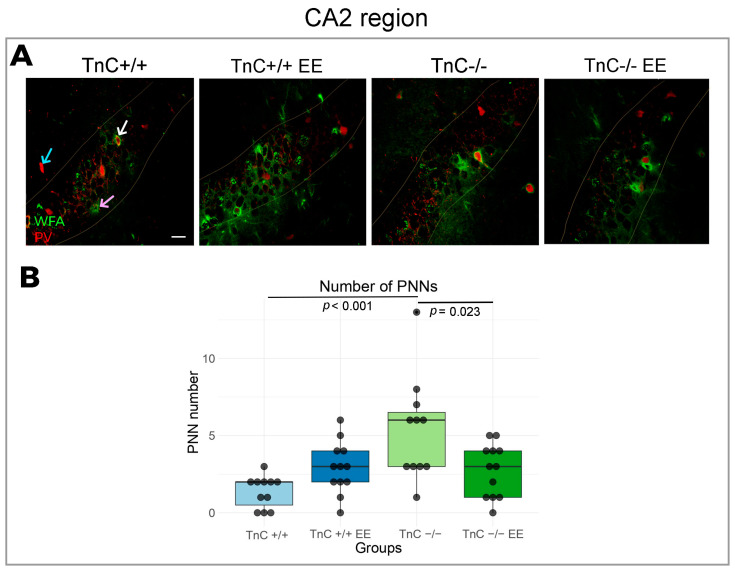
Expression of PNNs in the CA2 region in the hippocampus and the role of TnC and the exposure to EE. (**A**) Representative confocal images showing double immunofluorescence staining for the PNN marker WFA (green), and PV marker (red) showing PNN+PV+ cells (white arrow), PNN+PV- (pink arrow) and PV+PNN- (blue arrow). Pyramidal layer of the CA2 is delineated with light yellow line. (**B**) Box plots depicting the median number of PNNs (4 animals per experimental group, 4 brain section per animal); whiskers extending from the box indicate the range of data within 1.5 times the interquartile range. Significant differences were revealed in (**B**) PNN number upon TnC depletion (TnC-/-) and with EE in these mice (two-way ANOVA, *p* < 0.001; *p* = 0.023, respectively). Scale bar (**A**): 50 μm.

**Figure 5 biomolecules-14-00508-f005:**
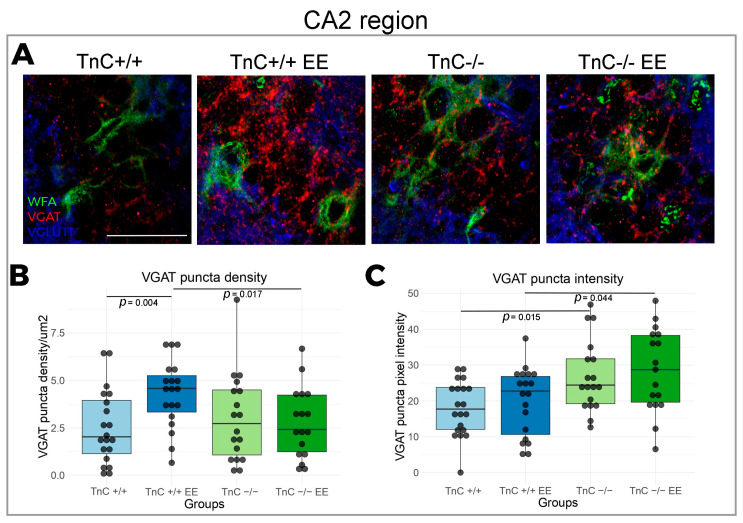
Expression of inhibitory and excitatory synapses in the CA2 region in the hippocampus and the role of TnC and the exposure to EE. (**A**) Representative 3D confocal images show triple immunofluorescent staining for the PNN marker WFA (green), the inhibitory synapse marker VGAT (red) and the excitatory synapse marker VGlut1 (blue). (**B**,**C**) Box plots depicting the density and intensity of inhibitory synapses that are penetrating PNN (3–4 animals per experimental group, 3 brain section per animal); whiskers extending from the boxes indicate the range of data within 1.5 times the interquartile range. Significant differences were revealed for (**B**) VGAT density with EE housing and for the role of TnC in this effect (Kruskal–Wallis, *p* = 0.004; *p* = 0.017, respectively) and (**C**) VGAT intensity for the role of TnC (TnC-/-) and EE in TnC deficiency (Kruskal–Wallis, *p* = 0.015; *p* = 0.044, respectively). Scale bar (**A**): 50 μm.

**Figure 6 biomolecules-14-00508-f006:**
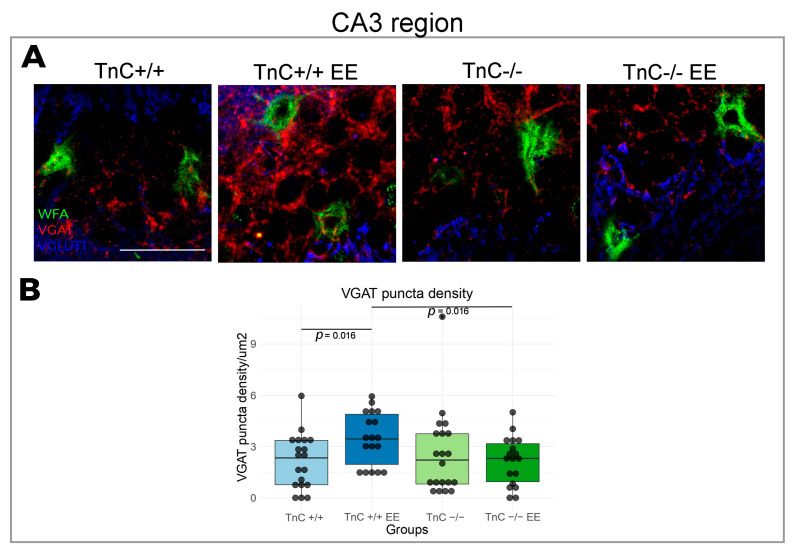
Expression of inhibitory and excitatory synapses in the CA3 region in the hippocampus, the role of TnC and the exposure to EE. (**A**) Representative 3D confocal images from show triple immunofluorescence staining for the PNN marker WFA (green), the inhibitory synaptic marker VGAT (red) and the excitatory synaptic marker VGlut1 (blue). (**B**) Box plot depicting the density of the inhibitory synapses that are penetrating the PNN (3–4 animals per experimental group, 3 brain section per animal); whiskers extending from the boxes indicate the range of data within 1.5 times the interquartile range. Significant differences were revealed in VGAT density for the effect of EE and the role of TnC in this effect (TnC-/- EE) (Kruskal–Wallis, *p* = 0.016; *p* = 0.016). Scale bar (**A**): 50 μm.

## Data Availability

The data that support the findings of this study are available from the corresponding authors upon reasonable request.

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
