# Peer review of "The Role of Tenascin-C on the Structural Plasticity of Perineuronal Nets and Synaptic Expression in the Hippocampus of Male Mice"

_biomolecules, 2024, doi:10.3390/biom14040508_

Round 1

Reviewer 1 Report

Comments and Suggestions for Authors

The manuscript presents compelling evidence concerning the influence of TnC deficiency on perineural networks within the hippocampus, along with an evaluation of the interaction between TnC expression and enriched environments. Overall, the manuscript is well-structured and presents clear and robust findings. However, there are somel concerns that need to be addressed before publication.

Firstly, the rationale behind the use of enriched environments is not adequately justified in the introduction section. Additionally, the inclusion of this variable introduces complexity in interpreting the obtained results. I recommend the authors provide a clearer justification for incorporating enriched environments, perhaps highlighting its relevance to the study's objectives and the existing literature.

Moreover, in the discussion and conclusion sections, it would be beneficial for the authors to offer more insight into the interaction between PNN-TnC-EE, elaborating on how each element may influence or modulate the others.

Regarding anatomical clarity, it's essential to specify the region of the hippocampus under investigation, whether it pertains to the dorsal or ventral region. Clear anatomical references are crucial for readers to accurately interpret the findings and understand the implications within the context of hippocampal subregions.

Comments on the Quality of English Language

None

Author Response

Firstly, the rationale behind the use of enriched environments is not adequately justified in the introduction section. Additionally, the inclusion of this variable introduces complexity in interpreting the obtained results. I recommend the authors provide a clearer justification for incorporating enriched environments, perhaps highlighting its relevance to the study's objectives and the existing literature.

Response: We thank the reviewer for this recommendation, we agree with the comment and are now providing a justification for incorporating enriched environment in the following lines 113-119:

“We previously demonstrated that EE in duration of 2 months increased the levels of TnC in the cerebellum [9]. Given that TnC expression declines with ageing, incorporating enriched environment represent a non-invasive approach to increase the brain plasticity together with upregulating expression of TnC. Moreover, in conditions of increased plasticity by EE, TnC acts as a modulatory molecule for PNN expression [9]. Besides affecting PNNs and TnC expression, EE leads to changes in the number of inhibitory and excitatory synaptic puncta in the cerebellum [9,66]”

Moreover, in the discussion and conclusion sections, it would be beneficial for the authors to offer more insight into the interaction between PNN-TnC-EE, elaborating on how each element may influence or modulate the others.

Response: We thank the reviewer for this comment and made elaboration of how each elemental factor may modulate the others in the following lines in discussion and conclusion:

Lines 471-475: “The interaction of the TnC deficient genotype and the EE resulted in distinct alterations of PNNs and synaptic expression in the DG. EE recovered the number of nodes in the PNN mesh and the excitatory input in TnC deficient mice. Considering this, EE could have a compensatory effect on TnC deficiency background. On the other hand, dynamic role of TnC in the DG might be dependent on the environmental context.”

Lines 506-510: “Moreover, the function of TnC as a regulator of synaptic integrity becomes more pronounced in the conditions of increased neuronal stimulation induced by EE. This effect is evident in the CA2 and CA3 regions, where EE, only in presence of TnC affects the PNN number in the CA2 region, and increases the inhibitory input in both regions.”

Lines 547-551: “The most prominent effects of TnC deficiency are seen in the DG and CA2 subfields, where TnC showed to be instrumental in decreasing synaptic plasticity by keeping the integrity of PNNs. Enriched environment may have additional effect on PNNs and synaptic marker expression, as it is increasing plasticity, which could rescue some of the changes caused by TnC deficiency.”     

Regarding anatomical clarity, it's essential to specify the region of the hippocampus under investigation, whether it pertains to the dorsal or ventral region. Clear anatomical references are crucial for readers to accurately interpret the findings and understand the implications within the context of hippocampal subregions.

Response: We thank the reviewer for this comment. In the abstract (line 23), methodology (lines 162, 202), results (line 298), discussion (line 534), and conclusion (line 547) it is specified that investigation was performed only on dorsal hippocampus.

Reviewer 2 Report

Comments and Suggestions for Authors

In this study, the authors investigated the role of TnC in the perineuronal network in the hippocampus using TnC KO mouse model. The authors reported a significant role of TnC in ultrastructures and the number of PNNs in DG and CA2 of the hippocampus.

This study could be interesting, however some points should be answered as follows:

Points

1. In this study, it seems that the experimental design is not properly designed. Therefore, the interpretation of the research results has various variables, so the correct interpretation of TNC has not been made. Overall, I think it is necessary to solve these problems and reconstruct the paper again.

2.  The description in the Abstract is described very elusive. The abstract should be revised to accurately describe the results. In particular, the following sentences are very difficult to understand. : 'TnC -/- mice in SE showed ultrastructural reduction of PNN mesh and increased inhibition in DG, and increase in the number of PNNs and inhibition in CA2 region, compared to TnC+/+ SE group. TnC+/+ mice after EE showed increased inhibition in the CA2, CA3 and DG region, com- 28 pared to TnC+/+ in SE, and in DG the change was followed by increased intensity of PNNs around PV+ neurons.'

Comments on the Quality of English Language

The English of this paper is difficult to understand, so I think it's better to get a revision from a native speaker again.

Author Response

  1. In this study, it seems that the experimental design is not properly designed. Therefore, the interpretation of the research results has various variables, so the correct interpretation of TNC has not been made. Overall, I think it is necessary to solve these problems and reconstruct the paper again.

Response: We thank the reviewer for this insight. As suggested, we made substantial changes and reconstructed the paper to properly address the role of TnC in shaping hippocampal plasticity. This can be found in the thoroughly revised Results and Discussion sections.         

  1. The description in the Abstract is described very elusive. The abstract should be revised to accurately describe the results. In particular, the following sentences are very difficult to understand. : 'TnC -/- mice in SE showed ultrastructural reduction of PNN mesh and increased inhibition in DG, and increase in the number of PNNs and inhibition in CA2 region, compared to TnC+/+ SE group. TnC+/+ mice after EE showed increased inhibition in the CA2, CA3 and DG region, com- 28 pared to TnC+/+ in SE, and in DG the change was followed by increased intensity of PNNs around PV+ neurons.'

Response: We thank the reviewer for pointing this out. We have rewrote the results in the Abstract in the following manner (lines 24-31):

“Deletion of TnC in TnC -/- mice showed an ultrastructural reduction of the PNN mesh and an increased inhibitory input in the dentate gyrus (DG), and an increase in the number of PNNs with a rise in the inhibitory input in the CA2 region. EE induced an increased inhibitory input in the CA2, CA3 and DG, while in DG the change was also followed by an increased intensity of PNNs. No changes in PNNs or synaptic expression were found in the CA1 region. We conclude that DG and CA2 regions emerged as focal points of alterations in PNNs and synaptogenesis with EE as mediated by TnC.”

Reviewer 3 Report

Comments and Suggestions for Authors

Major Comments:

1. Why was this study limited to Male mice only?

2. Authors concluded about affecting synaptic plasticity as an effect of TnC deletion. Is there any experiment done to conform these observations in animal behavior? A hippocampus dependent behavior task quantification should have been performed.

3. Why CA1 is different than other sub regions in this context? Any probable mechanism, synaptic input quantification done?

Minor Comment:

The image quality of boxplots in each figure should be improved.

Author Response

Reviewer 3:

  1. Why was this study limited to Male mice only?

Response: We appreciate the reviewer’s observation. Our decision to focus exclusively on male mice was made on one hand on our previous experience with the TnC model (see Stamenkovic et al, 2017, and Andjus et al, 2005), and on the other the limited resources. We agree with this comment, and stated in the abstract that results were based on male mice and acknowledge our limitation at the end of the Discussion:

Lines 539-543: “Of note, in our study we used only male mice based on comparison with previous experience with males in the TnC model [9,56]. Thus, further study of the expression and ultrastructure of PNNs with comparison of female and male mice, together with measurements of inhibition and excitation in the hippocampus of TnC deficient mice is warranted.”    

  1. Authors concluded about affecting synaptic plasticity as an effect of TnC deletion. Is there any experiment done to conform these observations in animal behavior? A hippocampus dependent behavior task quantification should have been performed.

Response: We thank the reviewer for pointing this out. We acknowledge the importance of hippocampus-dependent behavioral tasks in the TnC -/- mice in discovering its effect on synaptic plasticity. We did not include these aspects in our present study. However, we and others did previously perform behavioral testing in the TnC deficient model, as we explain now in following lines 85-94:  

“The first behavioral study using TnC deficient (TnC-/-) mice examined their ability to encode spatial information, and found that their performance in the water maze was unaltered [52]. However, in our previous study, we showed impaired spatial learning capabilities in TnC deficient mice, using Morris water maze test, albeit detectable only after being reinforced by housing in an enriched environment [53]. Next, Morellini and Schachner [54] further addressed hippocampal functions of TnC-/- mice, by showing impaired extinction of the conditional fear response and enhanced novelty-induced activity in the male mice deficient in TnC. We also reported enhanced novelty-induced locomotion and exploration activity on TnC deficient mice, which was stabilized by EE [53].”   

  1. Why CA1 is different than other sub regions in this context? Any probable mechanism, synaptic input quantification done?

Response: We thank the reviewer for the insightful comment. Knowing the differences in function of each hippocampal subregion, we propose the explanation in:

Lines 519-523: “The lack of the effect of TnC depletion in the CA1 region may reflect the absence of involvement of this biomolecule in associative memory and spatial navigation, the primary functions of the CA1 region. This assumption is supported by the previous study that demonstrated that spatial learning and memory were unaltered in TnC -/- mice [52].”

Lines 536-539: “TnC could be specifically involved in modulating structural and synaptic plasticity within the functional networks in the DG and CA2, which are responsible for pattern separation and social memory, respectively, without affecting primary functions of the CA1 and CA3 regions.”

Minor comment:       

The image quality of boxplots in each figure should be improved.

Response: The reviewer can now hopefully appreciate new better colored and emphasized boxplots in each relevant figure.

Round 2

Reviewer 3 Report

Comments and Suggestions for Authors

I am happy with the responses. I have no further comments.